# Comparative Effects of Dietary Hemp and Poppy Seed Oil on Lipid Metabolism and the Antioxidant Status in Lean and Obese Zucker Rats

**DOI:** 10.3390/molecules25122921

**Published:** 2020-06-25

**Authors:** Bartosz Fotschki, Paulina Opyd, Jerzy Juśkiewicz, Wiesław Wiczkowski, Adam Jurgoński

**Affiliations:** 1Department of Biological Function of Foods, Institute of Animal Reproduction and Food Research, Division of Food Science, Tuwima 10, 10-748 Olsztyn, Poland; b.fotschki@pan.olsztyn.pl (B.F.); p.opyd@pan.olsztyn.pl (P.O.); j.juskiewicz@pan.olsztyn.pl (J.J.); 2Department of Chemistry and Biodynamics of Food, Institute of Animal Reproduction and Food Research, Division of Food Science, Tuwima 10, 10-748 Olsztyn, Poland; w.wiczkowski@pan.olsztyn.pl

**Keywords:** unsaturated fatty acids, obesity, lipid profile, liver disorders, oxidative stress, Zucker rats

## Abstract

The objective of this study was to compare the effects of the dietary inclusion of hemp seed oil (HO) and poppy seed oil (PO) on the lipid metabolism and antioxidant status of lean and genetically obese Zucker rats. The rats were fed a control diet for laboratory rodents or a modification with HO or PO. Both oils reduced body and epididymal fat and liver cholesterol levels and promoted oxidative stress in the liver of obese rats. The HO reduced plasma triglycerides and had a stronger liver cholesterol-lowering effect in obese rats than PO. In the lean rats, HO and PO had no effects on the body fat content, plasma lipid profile, or lipid metabolism in the liver. HO considerably elevated the content of α-linolenic acid in the liver and increased the liver ratio of reduced glutathione (GSH)/oxidized glutathione (GSSG) in the lean rats. In conclusion, the regular consumption of both oils increases the accumulation of essential fatty acids in the liver of healthy animals, whilst not having any adverse effects on the body, whereas in genetically obese rats, the effects of both dietary oils on the lipid metabolism and antioxidant status are unequivocal and only partially beneficial.

## 1. Introduction

Poppy and hemp are plant species mainly associated with their psychoactive and pain relief effects. However, their seeds are also a good source of nutrients and other components, the consumption of which can be beneficial for health. These include fat, which comprises approximately 30% and 50% of hemp and poppy seeds, respectively [1,2,3]. Oils extracted from hemp and poppy seeds are rich in polyunsaturated fatty acids (PUFAs), especially essential fatty acids, such as omega-6 linoleic acid and omega-3 α-linolenic acid, although their composition considerably differs between one another. Primarily, poppy seed oil has a high percentage of linoleic acid, just like sunflower oil (approx. 70% of total fatty acids (TFAs)), and only trace amounts of α-linolenic acid (up to 1%) [3,4]. On the other hand, hemp seed oil, besides displaying considerable amounts of linoleic acid (up to 56% of TFAs), is also abundant in α-linolenic acid (20% of TFAs, which is over twice as much as in canola-type rapeseed oil). In addition, hemp seed oil contains small percentages of other PUFAs, such as γ-linolenic acid and stearidonic acid, which are rarely seen in vegetable oils [5]. Poppy and hemp seed oils are also a source of antioxidative vitamins, especially tocopherols, and other bioactive compounds, such as phenolics and phytosterols, including β-sitosterol and campesterol [3,6]. Recent studies on these oils have been focused on their quality and stability and potential use in the food industry [7,8], whereas their health effects, especially the effects of poppy seed oil, have not been extensively investigated thus far [9,10].

Obesity is a complex genetically-related condition involving the dysregulation of several organ systems and molecular pathways, including the liver, gastrointestinal tract, microbiome, and central nervous system. One solution to prevent or mitigate obesity and its related disorders is a well-balanced diet [11]. Among dietary constituents, fat has gained the most recognition in affecting consumer health. Diets enriched with saturated and trans fatty acids have been implicated in obesity and heart disease, whereas PUFAs are mostly associated with beneficial effects on health [12,13]. The omega-6 and omega-3 fatty acids play an important role in health and disease by generating potent modulatory molecules regulating oxidative stress and immune-metabolic disorders, improving the blood lipid profile and blood pressure [14,15,16,17,18]. However, there are also some controversies regarding the bioactivity of PUFAs. For example, a dietary overabundance of long-chain omega-3 PUFAs may trigger oxidative stress, whereas the overconsumption of linoleic acid may promote inflammation [19,20]. These fatty acids are incorporated into the cell membranes and lipoproteins, which may lead to an increased unsaturation index and thus induce higher lipid peroxidation [21]. Moreover, a potential difference in the effects of PUFA supplementation on a healthy organism and an organism struggling with metabolic disorders remains an open and current question [21,22].

Taking into account the growing popularity of oils from unconventional sources in the food market, hemp and poppy seed oils are an interesting alternative for common vegetable oils, so potential benefits resulting from their consumption need to be clarified in different conditions. The objective of this study was to compare the effects of the dietary inclusion of hemp seed oil and poppy seed oil on the lipid metabolism and antioxidant status of lean and genetically obese Zucker rats. We hypothesized that dietary hemp seed oil and poppy seed oil can differentially affect the lipid metabolism and antioxidant status, depending on the rat phenotype and oil source.

## 2. Results

After the 4-week feeding period, the growth-related parameters were considerably affected when comparing the lean group to the obese groups (*p* < 0.001, Table 1). The diet intake, body weight, body fat, and epididymal fat were considerably increased in the obese groups compared to those of the lean groups. Additionally, in these groups, lower body lean values were also observed (*p* < 0.001). Some observed obese-related disorders were mitigated when hemp or poppy seed oil was used as the source of dietary fat. Both oils significantly reduced the body fat percentage and increased the body lean percentage in obese rats (*p* < 0.05). Among the obese groups, the strongest reduction in the percentage of epididymal fat was observed when poppy seed oil was added to the diet (*p* < 0.05). In the lean rats, only an elevation of body lean percentage was found in the group fed hemp seed oil (*p* < 0.001).

The plasma lipid profile and antioxidant status were affected by the tested factors and are shown in Table 2 and Figure 1. When compared to lean rats, the concentrations of total cholesterol, high-density lipoprotein cholesterol (HDL), low-density lipoprotein cholesterol (LDL), and triglycerides were increased in obese rats (*p* < 0.001). Moreover, in obese groups, the plasma levels of the antioxidant capacity of water-soluble substances (ACW) were elevated compared to those of lean groups (*p* < 0.001). Dietary poppy seed oil undesirably increased LDL cholesterol and reduced HDL cholesterol in obese rats (*p* < 0.005). Dietary hemp seed oil, in turn, partially mitigated obese-related lipid disorders through a reduction of triglyceride concentrations in plasma (*p* < 0.005), which was not observed in lean rats (interaction at *p* < 0.05). Nevertheless, the diet with hemp seed oil unfavourably reduced the concentration of HDL cholesterol and the ACW level in obese rats. Both examined oils had no significant effect on the plasma lipid profile or antioxidant status indicators in the lean rats.

Considerable disorders of the lipid metabolism and antioxidant status were observed in the liver (Table 3 and Figure 2). The two-way ANOVA showed a significantly higher liver weight and GSH/GSSG ratio (*p* < 0.05), as well as significantly higher liver fat, cholesterol, and triglyceride contents (*p* < 0.001), in obese rats compared to lean rats. Furthermore, in obese rats, the hepatic contents of α-linolenic, linoleic, and oleic acid were significantly reduced compared to those in lean rats (*p* < 0.001). The observed changes in the liver of obese rats were partially regulated by the addition of poppy or hemp seed oil to the diet. Both oils considerably decreased the concentration of liver cholesterol (*p* < 0.01); however, the strongest effect was observed when obese rats were fed a diet with hemp seed oil (*p* ≤ 0.05). The concentrations of α-linolenic and linoleic acid in the livers of obese rats were not affected by the tested oils. Nevertheless, in lean rats fed a diet with hemp seed oil, the hepatic concentrations of these acids were considerably elevated compared to those of lean rats fed the control diet (*p* ≤ 0.05). In the case of poppy seed oil, a significantly higher concentration of linoleic acid was observed in the liver of lean rats than in the liver of lean rats fed the control diet (*p* ≤ 0.05). The analyses of the liver antioxidant status of obese rats showed that diets with hemp and poppy seed oil elevated oxidative stress to a different extent by increasing the malondialdehyde (MDA) values (OPO > OHO, *p* ≤ 0.05) and by decreasing the reduced glutathione (GSH)/oxidized glutathione (GSSG) ratios (*p* ≤ 0.05). The opposite effect (GSH/GSSG ratio) or lack of the effect (MDA content) was observed when lean rats were fed the oils (interactions at *p* < 0.01 and *p* < 0.005, respectively). More specifically, the post hoc statistical analysis indicated that in the LHO group, the liver GSH/GSSG ratio was favourably increased compared to that of the lean control group (*p* ≤ 0.05).

## 3. Discussion

We aimed to compare the effects of the dietary inclusion of hemp seed oil and poppy seed oil on the lipid metabolism and antioxidant status of lean and genetically obese rats. Therefore, the first step of our study was to determine the fatty acid composition of the tested oils. They were both rich in PUFAs, the total content of which was 68% and 76% of TFAs for the poppy seed oil and hemp seed oil, respectively (Table 1). In the case of the poppy seed oil, PUFAs consisted almost solely of linoleic acid (67% of TFAs), whereas the hemp seed oil contained both a high percentage of linolenic acid (52.8% of TFAs) and α-linolenic acid (17.5% of TFAs). Therefore, its omega-6/omega-3 ratio was 3.3, which is a very low value and can hardly be found in vegetable oils. It is worth noting that an increase in the omega-6/omega-3 ratio is associated with an increased risk of obesity in men [12]. Nevertheless, our results are similar to those obtained by other authors who determined the fatty acid profile of the oils [5,8]. However, in the aforementioned studies, the contents of linoleic and α-linolenic acids differed by 2–4% and 0.3–1%, respectively, which may be a consequence of the various growth conditions of hemps and poppies and/or the extraction method that was applied. The hemp seed oil also contained γ-linolenic acid (4.3% of TFAs), which has been recognized as having anti-inflammatory and anticancer activities [14]. Interestingly, the hemp seed oil also contained a small amount of trans fatty acid (*cis*-9, *trans*-12- octadecadienoic acid; 0.12% of TFAs). Trans fatty acids are generally recognized as detrimental to the body because they raise the risk of cardiovascular disease by inhibiting the synthesis of PUFAs in arterial cells [23]. Moreover, the tested hemp and poppy seed oils contained similar amounts of saturated fatty acids (SFAs) (9% and 10% of TFAs, respectively), whereas the former was richer in monounsaturated fatty acids (MUFAs) than the latter (17% and 10% of TFAs, respectively).

Experimental studies have suggested that diets enriched with omega-3 and omega-6 fatty acids may modulate body fat gain through changes in the lipid metabolism and inflammation processes [12]. Indeed, in this study, the obese rats fed diets with hemp and poppy seed oils had considerably lowered final body fat and epididymal fat contents. Moreover, in these groups, a beneficial elevation of the final body lean mass was observed. Rosqvist et al. [24] observed that diets enriched with PUFAs elevated equal amounts of fat and lean tissue, but diets enriched with SFAs increased the fat tissue four times as much as the lean tissue in men. Cardel et al. [25], in a cohort study on healthy, racially diverse children aged 7–12 years, presented correlations between a higher consumption of PUFAs and increased values of body lean mass, as well as a reduction of body fat mass. The mechanisms underlying the differences in body fat content might be related to the different effects of fatty acids on lipogenesis. Dietary PUFAs may suppress hepatic lipogenesis through mechanisms regulated by sterol regulatory element-binding proteins or by the suppression of fatty acid β-oxidation through PPARα [26,27].

Previous studies have shown a clear association between an increased body weight, obesity, and a higher synthesis of cholesterol [28]. Additionally, in the present study, the concentration of liver cholesterol was considerably increased in the obese rats, clearly showing that hemp seed oil and, to a lesser extent, poppy seed oil, lowered the concentration of liver cholesterol. This effect might be linked to a higher content of PUFAs, especially omega-3 fatty acids in the hemp seed oil. Nutritional studies on rats and mice have shown that diets supplemented with omega-3 fatty acids induce changes in the transport and metabolic pathways of cholesterol in the liver, resulting in a more rapid disposition of plasma-derived cholesterol into the bile [29,30]. It is worth emphasizing that the cholesterol-lowering effects of both oils were only seen in obese rat livers. This difference is most likely associated with a considerably higher feed consumption in obese rats than lean rats, which was cholesterol-free, thus enhancing the effect of dietary oils on cholesterol metabolism.

It is well-known that a diet supplemented with omega-3 fatty acids favorably modulates the blood lipid profile [31]. In our study, the effects of dietary hemp seed oil on the blood lipid profile were unequivocal and depended on the rat phenotype, whereas the effects of the poppy seed oil were rather negative, but only in obese rats. Dietary hemp seed oil markedly reduced the concentration of plasma triglycerides in obese rats to the levels observed in the lean rats. The significant regulatory effect of hemp seed oil on plasma triglycerides, besides being related to a high amount of omega-3 fatty acids, might also be associated with the presence of omega-6 γ-linolenic acid, which can regulate plasma triglyceride levels [32]. Nevertheless, both oils had unfavorable effects in obese rats, as exhibited by the reduction of plasma HDL cholesterol levels. Additionally, poppy seed oil elevated plasma LDL cholesterol levels. The undesired effects of the poppy seed oil on the blood lipid profile might partially be a result of the high omega-6/3 ratio, which was 66. A high omega-6/3 ratio in a diet is associated with increased concentrations of serum LDL cholesterol and triglycerides [33]. Conversely, the hemp seed oil contained trans fatty acids, which are known to have undesirable effects on the blood lipid profile, including a reduction of plasma HDL cholesterol levels [34]. Another important factor in the modulation of the lipid profile might have been the rat phenotype. In our previous study on obese Zucker rats, a reduction of the plasma HDL cholesterol level after dietary supplementation with hemp seed oil was also observed [35]. Interestingly, changes in the blood lipid profile were not the case in healthy animals, which confirms our hypothesis that the health state of rats is an important factor in the modulation of lipid metabolism by the tested oils. The lean rats did not struggle with metabolic disorders, so their organisms were probably able to maintain metabolic homeostasis, despite dietary changes. Nevertheless, the lipid metabolism between rats and humans differs. Therefore, human trials on the effects of dietary supplementation with hemp and poppy seed oils are necessary to confirm the results obtained here.

In this study, both dietary oils, but to a different extent, induced oxidative stress in the liver of obese rats through the elevation of a by-product of lipid peroxidation (MDA) and/or the reduction of the GSH/GSSG ratio. Additionally, dietary hemp seed oil also unfavorably decreased the plasma ACW level in obese rats. However, all of these unfavorable changes were not seen in healthy animals. Therefore, we speculate that the induction of oxidative stress might be due to overloaded hepatocytes with fat and simultaneous PUFA oxidation, which are the substrates required for MDA production [36]. Indeed, an increased concentration of linoleic acid by both oils and increased concentration of α-linolenic acid by dietary hemp seed oil were only seen in lean animals, whereas in the liver of obese rats, the concentrations were comparable among all groups. However, in the lean rats, dietary hemp seed oil was even able to favorably increase the hepatic ratio of GSH/GSSG, suggesting that PUFAs can have positive effects on the antioxidant status, if not disturbed by an obesogenic state. This seems to be in agreement with Firat et al. [37], who observed an inhibition of oxidative stress in rat livers by omega-3 fatty acids.

## 4. Materials and Methods

### 4.1. Chemical Composition of Oils

Unrefined, cold-pressed industrial hemp (*Cannabis sativa* L.) and poppy (*Papaver somniferum* L.) seed oils were obtained from the Ol’Vita company (Panków, Poland). Palm oil was obtained from Vog Polska Ltd. (Skierniewice, Poland). The fatty acid profile of the oils was determined using gas chromatography with flame-ionization detection after the conversion of the fatty acids into methyl esters according to a previously described method [38]. The fatty acid profiles of the hemp and poppy seed oils are shown in Table 4.

### 4.2. Animal Study

This study was carried out in strict accordance with the recommendations of the National Ethics Commission (Warsaw, Poland). All procedures and experiments complied with the guidelines and were approved by the Local Ethics Commission of the University of Warmia and Mazury (Olsztyn, Poland, Permit Number: 37/2017) with respect to animal experimentation and the care of animals under study, and all efforts were made to minimize suffering. To examine the effects of oil consumption on healthy and obese organisms, 7-week-old Zucker rats were divided into three groups (eight animals each) of the lean phenotype (L) and three groups (six animals each) of the obese phenotype (O). The Zucker rats used as a genetic model of obesity were obtained from Charles River Laboratories, Research Models and Services, Germany GmbH. Animals were housed individually in the cages with wire mesh floors to prevent coprophagia. The initial body weights were comparable among the lean and obese groups and were 148 ± 1.1 and 185 ± 5 g, respectively. For 4 weeks, each group was fed a modified version of the semi-purified rodent diet AIN-93G recommended by Reeves [39]. All experimental diets were similar in terms of dietary ingredients, except for the type of oil. The Zucker rats from lean (LC) and obese (OC) control groups were fed a diet containing palm oil as the sole source of fat (7% of diet), whereas in the experimental groups, palm oil was replaced with hemp seed oil (groups LHO and OHO) or poppy seed oil (groups LPO and OPO). Unlike the tested oils, palm oil contained considerable amounts of saturated and monounsaturated fatty acids (SFAs and MUFAs). Both palm oil in the control diet and its complete replacement with the tested oils were applied in order to obtain the most effective response of the organism to experimental factors, so that differences between poppy and hemp seed oils in terms of their health effects would be as clear as possible. Details about the proportional composition of each group-specific diet are shown in Table 5. Animals were fed a fresh diet everyday ad libitum with continuous access to water. The individual body weights of rats and their food intake were recorded on a weekly and daily basis, respectively. The animals were maintained under standard conditions at a temperature of 21–22 °C and a relative air humidity of 50–70% with intensive room ventilation (15×/h) and a 12 h lighting regimen.

### 4.3. Sample Collection, Growth Parameters, and Body Composition

Experimental groups were monitored for feed intake, body weight, fat content, and lean mass using an NMR analyser (nuclear magnetic analyser, Minispec LF110, BRUKER, Karlsruhe, Germany). At the end of the experiment, the rats were weighed and anaesthetized i.p. with ketamine (K) and xylazine (X) (K, 100/kg BW; X, 10 mg/kg BW), according to the recommendations for the anaesthesia of experimental animals. After a laparotomy, blood samples were collected from the caudal vein and stored in tubes containing heparin. Plasma was prepared by low-speed centrifugation (350× *g*, 10 min, 4 °C) and kept frozen at −80 °C until assayed. The epididymal fat and the liver were removed and weighed.

### 4.4. Liver Antioxidant Status and Lipid Content

Malondialdehyde (MDA) was determined in the liver tissue after storage at −80 °C. A procedure developed by Botsoglou et al. [40] was used in the assay, and the MDA contents were determined spectrophotometrically at 532 nm and expressed in µg malondialdehyde per g of tissue. In the liver tissue, the reduced glutathione (GSH) and oxidized glutathione (GSSG) concentrations were determined by using the enzymatic recycling method described by Rahman et al. [41]. Liver lipids were extracted according to Folch et al. [42]. Following extraction, the fat content, total cholesterol (TC), and triglyceride (TG) concentrations were determined enzymatically using commercial kits (Cholesterol DST, Triglycerides DST, Alpha Diagnostics, Ltd., San Antonio, TX, USA). The contents of α-linolenic acid, linoleic acid, and oleic acid in the liver were determined according to the modified method of Bromke et al. [43]. Dried samples obtained from Folch’s extraction were mixed with 1 mL of methanol/methyl tert-butyl ether/water (1/3/1, *v/v/v*). After extraction in an ultrasound bath (3 min, 4 °C), 0.5 mL of a mixture of methanol/water (1/3, *v/v*) was added and vortexed for 30 s, and the formed lipid phase was collected. The step was repeated two times, and the collected phases were combined and then evaporated to dryness under a nitrogen atmosphere. The obtained residue was dissolved in 0.2 mL of methanol/6% KOH (4/1, *v/v*), and 0.1 mL of NaCl (saturated solution) and 0.05 mL of 29% HCl were then added. The fatty acids were extracted three times with 2 mL of chloroform/heptanes (1/4, *v/v*) by sonication and vortexing (3 × 30 s), and the collected organic phases were combined, evaporated to dryness under a nitrogen atmosphere, and stored at −80 °C until analysis. Before analysis, the samples were dissolved in 0.1 mL of methanol and centrifuged (4 °C, 13,200× *g*, 20 min). Determination of the profile and content of fatty acids was carried out using a micro-HPLC-MS/MS system (Sciex, Redwood, CA, USA) consisting of a dual-channel pump, column oven, autosampler, system controller, and 5600 QTOF mass spectrometer. Chromatographic separation was conducted on a HALO C18 column (2.7 µm, 0.5 mm × 50 mm; Eksigent, Dublin, Ohio, USA) at 60 °C and with a flow rate of 15 µL/min. The elution solvents A (water/0.5 M ammonium formate/formic acid, 98.9/1/0.1, *v/v/v/v*) and B (acetonitrile/0.5 M ammonium formate/isopropanol/formic acid, 68.9/1/30/0.1, *v/v/v/v*) were used in the following gradient schema: 30-30-90-90-30-30% A from 0-0.5-1.8-2.8-3-5 min. The MS QTOF conditions were as follows: negative ionization; nitrogen curtain gas, 25 L/min; ion spray source voltage, −4500 V; temperature, 350 °C; nebulizer gas 1, 35 L/min; turbo gas, 30 L/min; (Q1/Q2) DP, −90/−100 V; (Q1/Q2) CE, −25/−50 V; and CES, 15 V. Fatty acids were identified based on retention time-matching with available standards and the MS/MS spectra.

### 4.5. Plasma Antioxidant Status, Lipid Profile, and Inflammatory Indicators

In the blood plasma, the antioxidant capacity of water-soluble and lipid-soluble substances (ACW and ACL, respectively) was determined by a photochemiluminescence detection method using a Photochem spectrometer and respective kits (ACW-Kit and ACL-Kit, Analytik Jena AG, Jena, Germany). In the photochemiluminescence assay, the generation of free radicals was partially eliminated through reactions with antioxidants present in the plasma samples, and the remaining radicals were quantified by luminescence generation. Ascorbate and Trolox calibration curves were used to evaluate ACW and ACL, respectively. The TG, TC, fractions of HDL and LDL cholesterol, aspartate transaminase (AST), and alanine transaminase (ALT) concentrations were estimated using a biochemical analyser (Pentra C200, Horiba, Tokyo, Japan).

### 4.6. Statistical Analysis

STATISTICA software (version 10.0; StatSoft Corp., Kraków, Poland) was used to determine whether variables differed among the treatment groups. Two-way ANOVA was applied to assess the effects of the oil type (hemp seed oil or poppy seed oil; O), the Zucker rat phenotype (lean or obese; T), and the P-interaction between these investigated factors (O × T). If the ANOVA revealed a significant P-interaction (*p* < 0.05), the differences between the individual groups were then assessed with Duncan’s multiple range post hoc test at *p* < 0.05. The results are presented as the mean values ± standard error of the means (SEM), except for the fatty acid contents of the examined oils, which are expressed as the mean ± the standard deviation (SD) of the mean.

## 5. Conclusions

In summary, this study shows that hemp and poppy seed oil are valuable dietary sources of PUFAs and their regular consumption can increase the accumulation of essential fatty acids in the liver of healthy animals and, at the same time, does not cause any adverse effects to the body. By contrast, in a genetically determined obesogenic state, the effects of both dietary oils on the lipid metabolism and antioxidant status are unequivocal and only partially beneficial. Both oils reduced the body fat, epididymal fat, and liver cholesterol levels in obese rats, whereas the hemp seed oil was additionally able to prevent a considerable increase in the plasma triglyceride levels and had stronger liver cholesterol-lowering effects. At the same time, the blood cholesterol profile was disturbed by both oils, especially by the poppy seed oil. Moreover, it was found that both examined oils promoted oxidative stress in obese rats, which apparently was a result of intensified PUFA oxidation in the liver. In conclusion, this study shows that the genetically determined health state of rats is an important factor affecting the modulation of lipid metabolism and the antioxidant status by the tested oils.

## Figures and Tables

**Figure 1 molecules-25-02921-f001:**
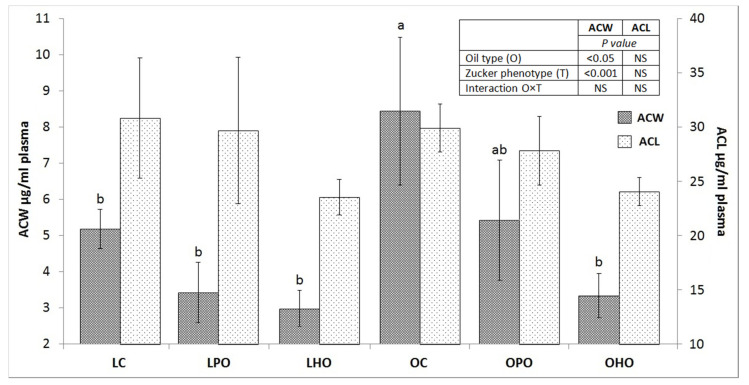
Plasma ACW and ACL levels in Zucker rats fed experimental diets. Values are the mean ± SEM, where *n* = 6 for obese rats and *n* = 8 for lean rats. Means without a common letter differ, *p* ≤ 0.05. LC, lean rats fed a control diet with palm oil; LPO, lean rats fed a diet with poppy seed oil; LHO, lean rats fed a diet with hemp seed oil; OC, obese rats fed a control diet with palm oil; OPO, obese rats fed a diet with poppy seed oil; OHO, obese rats fed a diet with hemp seed oil; ACW, antioxidant capacity of water-soluble substances; ACL, antioxidant capacity of lipid-soluble substances.

**Figure 2 molecules-25-02921-f002:**
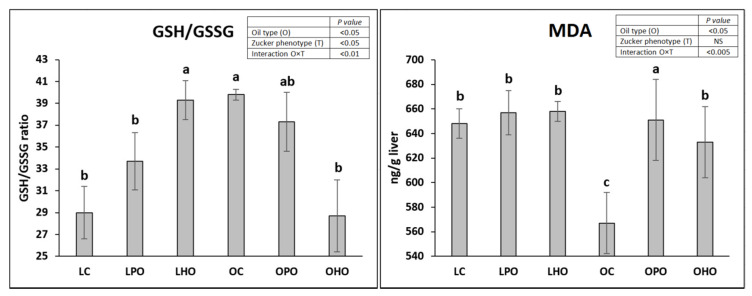
Liver antioxidant status in Zucker rats fed experimental diets. Values are the mean ± SEM, where *n* = 6 for obese rats and *n* = 8 for lean rats. Means without a common letter differ, *p* ≤ 0.05. LC, lean rats fed a control diet with palm oil; LPO, lean rats fed a diet with poppy seed oil; LHO, lean rats fed a diet with hemp seed oil; OC, obese rats fed a control diet with palm oil; OPO, obese rats fed a diet with poppy seed oil; OHO, obese rats fed a diet with hemp seed oil; GSH/GSSG, reduced glutathione to oxidized glutathione ratio; MDA, malondialdehyde.

**Table 1 molecules-25-02921-t001:** Diet intake, body weight, and body composition of rats after 4 weeks of feeding with experimental diets.

	Diet Intake	Body Weight	Body Weight Gain	Body Lean	Body Fat	Epididymal Fat
	g/day	g	g	%	%	%
Group ^1^						
LC (*n* = 8)	15.6 ± 0.3	259 ± 3.4	113 ± 3.2	66.4 ± 1.53 ^b^	23.6 ± 1.46 ^c^	0.96 ± 0.07 ^c^
LPO (*n* = 8)	15.1 ± 0.3	266 ± 4.5	117 ± 4.1	68.4 ± 0.59 ^ab^	23.2 ± 0.67 ^c^	0.86 ± 0.06 ^c^
LHO (*n* = 8)	15.0 ± 0.3	264 ± 5.1	115 ± 5.1	70.9 ± 0.92 ^a^	21.2 ± 0.82 ^c^	0.80 ± 0.05 ^c^
OC (*n* = 6)	24.2 ± 0.8	379 ± 14.2	197 ± 6.2	23.4 ± 1.36 ^d^	69.1 ± 1.25 ^a^	2.10 ± 0.08 ^a^
OPO (*n* = 6)	23.2 ± 0.9	377 ± 13.7	187 ± 7.4	29.9 ± 1.20 ^c^	65.1 ± 1.44 ^b^	1.91 ± 0.05 ^b^
OHO (*n* = 6)	24.2 ± 0.5	381 ± 13.2	197 ± 5.4	31.2 ± 1.00 ^c^	62.6 ± 0.81 ^b^	1.92 ± 0.09 ^ab^
Oil type (O)						
Control (*n* = 14)	19.3	310	149	48.0	43.1	1.45
Poppy seed oil (*n* = 14)	18.5	313	147	51.9	41.1	1.31
Hemp seed oil (*n* = 14)	18.9	314	150	53.9	38.9	1.23
*p value*	0.053	NS	NS	<0.001	<0.005	<0.05
Zucker phenotype (T)						
Lean (*n* = 24)	15.2	263	115	68.6	22.7	0.876
Obese (*n* = 18)	23.9	379	194	28.2	65.6	1.98
*p value*	<0.001	<0.001	<0.001	<0.001	<0.001	<0.001
Interaction O × T						
*p value*	NS	NS	NS	NS	NS	NS

Values are the mean ± SEM. NS, *p* > 0.05. ^1^ LC, lean rats fed a control diet with palm oil; LPO, lean rats fed a diet with poppy seed oil; LHO, lean rats fed a diet with hemp seed oil; OC, obese rats fed a control diet with palm oil; OPO, obese rats fed a diet with poppy seed oil; OHO, obese rats fed a diet with hemp seed oil. ^a–c^ Mean values within a column with different superscript letters were shown to be significantly different (*p* < 0.05); differences among the groups are only indicated with superscripts in the case of a statistically significant interaction O × T and differences between the oil type (*p* ≤ 0.05).

**Table 2 molecules-25-02921-t002:** Plasma lipid profile of rats fed experimental diets.

	TC	HDL	LDL	TG
	mmol/L	mmol/L	mmol/L	mmol/L
Group ^1^				
LC (*n* = 8)	3.15 ± 0.19	0.97 ± 0.12 ^c^	0.16 ± 0.02 ^c^	2.86 ± 0.54 ^b^
LPO (*n* = 8)	2.87 ± 0.21	0.86 ± 0.03 ^c^	0.22 ± 0.01 ^c^	2.52 ± 0.43 ^b^
LHO (*n* = 8)	2.81 ± 0.12	0.76 ± 0.09 ^c^	0.21 ± 0.02 ^c^	2.04 ± 0.29 ^b^
OC (*n* = 6)	5.66 ± 0.12	2.03 ± 0.04 ^a^	0.35 ± 0.03 ^b^	8.81 ± 1.62 ^a^
OPO (*n* = 6)	6.05 ± 0.32	1.52 ± 0.06 ^b^	0.54 ± 0.04 ^a^	9.10 ± 1.62 ^a^
OHO (*n* = 6)	5.47 ± 0.16	1.46 ± 0.02 ^b^	0.37 ± 0.02 ^b^	3.73 ± 1.08 ^b^
Oil type (O)				
Control (*n* = 14)	4.23	1.43	0.24	5.41
Poppy seed oil (*n* = 14)	4.34	1.14	0.36	5.56
Hemp seed oil (*n* = 14)	3.95	1.06	0.28	2.76
*p value*	NS	<0.005	<0.005	<0.005
Zucker phenotype (T)				
Lean (*n* = 24)	2.95	0.86	0.19	2.47
Obese (*n* = 18)	5.73	1.67	0.42	7.21
*p value*	<0.001	<0.001	<0.001	<0.001
Interaction O × T				
*p value*	NS	NS	NS	<0.05

Values are the mean ± SEM. HDL, high-density lipoprotein; LDL, low-density lipoprotein; NS, *p* > 0.05; TC, total cholesterol; TG, triglycerides. ^1^ LC, lean rats fed a control diet with palm oil; LPO, lean rats fed a diet with poppy seed oil; LHO, lean rats fed a diet with hemp seed oil; OC, obese rats fed a control diet with palm oil; OPO, obese rats fed a diet with poppy seed oil; OHO, obese rats fed a diet with hemp seed oil. ^a–c^ Mean values within a column with different superscript letters were shown to be significantly different (*p* < 0.05); differences among the groups are only indicated with superscripts in the case of a statistically significant interaction O × T and differences between the oil type (*p* ≤ 0.05).

**Table 3 molecules-25-02921-t003:** Hepatic lipid accumulation in rats fed experimental diets.

	Liver Mass	Fat Content	Cholesterol	TG	α-Linolenic Acid (omega-3)	Linoleic Acid (omega-6)	Oleic Acid (omega-9)
	g/100 g BW	%	mg/g liver	mg/g liver	mg/g liver		
Group ^1^							
LC (*n* = 8)	4.80 ± 0.03	7.36 ± 0.50	1.15 ± 0.02 ^d^	6.62 ± 0.72	0.07 ± 0.01 ^b^	8.03 ± 0.55 ^b^	8.51 ± 0.43
LPO (*n* = 8)	4.82 ± 0.14	7.04 ± 0.28	1.26 ± 0.05 ^d^	5.81 ± 0.82	0.28 ± 0.02 ^b^	21.49 ± 1.11 ^a^	9.56 ± 0.27
LHO (*n* = 8)	4.91 ± 0.06	7.35 ± 0.43	1.17 ± 0.05 ^d^	4.30 ± 0.52	2.13 ± 0.11 ^a^	21.13 ± 0.63 ^a^	8.64 ± 0.18
OC (*n* = 6)	5.53 ± 0.45	23.3 ± 1.98	1.87 ± 0.02 ^a^	25.7 ± 2.37	0.12 ± 0.01 ^b^	6.36 ± 0.45 ^b^	5.84 ± 0.39
OPO (*n* = 6)	5.92 ± 0.46	23.2 ± 2.87	1.67 ± 0.09 ^b^	26.0 ± 3.35	0.16 ± 0.01 ^b^	8.74 ± 0.54 ^b^	5.66 ± 0.25
OHO (*n* = 6)	6.05 ± 0.34	26.0 ± 2.93	1.45 ± 0.02 ^c^	25.9 ± 1.99	0.45 ± 0.04 ^b^	7.31 ± 0.71 ^b^	6.08 ± 0.44
Oil type (O)							
Control (*n* = 14)	5.11	14.22	1.46	14.81	0.10	7.20	7.18
Poppy seed oil (*n* = 14)	5.29	14.01	1.44	14.52	0.22	15.12	7.62
Hemp seed oil (*n* = 14)	5.40	15.42	1.29	13.62	1.29	14.22	7.37
*p value*	NS	NS	<0.01	NS	<0.001	<0.001	NS
Zucker phenotype (T)							
Lean (*n* = 24)	4.84	7.25	1.19	5.57	0.83	16.89	8.91
Obese (*n* = 18)	5.83	24.21	1.66	25.91	0.25	7.47	5.87
*p value*	<0.05	<0.001	<0.001	<0.001	<0.001	<0.001	<0.001
Interaction O × T							
*p value*	NS	NS	<0.001	NS	<0.01	<0.001	NS

Values are the mean ± SEM. NS, *p* > 0.05; TG, triglycerides. ^1^ LC, lean rats fed a control diet with palm oil; LPO, lean rats fed a diet with poppy seed oil; LHO, lean rats fed a diet with hemp seed oil; OC, obese rats fed a control diet with palm oil; OPO, obese rats fed a diet with poppy seed oil; OHO, obese rats fed a diet with hemp seed oil. ^a–d^ Mean values within a column with different superscript letters were shown to be significantly different (*p* < 0.05); differences among the groups are only indicated with superscripts in the case of a statistically significant interaction O × T and differences between the oil type (*p* ≤ 0.05).

**Table 4 molecules-25-02921-t004:** Fatty acid profile (%).

	Palm Oil	Poppy Seed Oil	Hemp Seed Oil
Myristic acid (C14:0)	0.74 ± 0.01	-	-
Palmitic acid (C16:0)	40.8 ± 0.04	8.35 ± 0.02	5.46 ± 0.01
Palmitoleic acid (C16:1)	0.12 ± 0.01	0.10 ± 0.00	0.08 ± 0.01
Stearic acid (C18:0)	4.39 ± 0.01	1.96 ± 0.01	2.26 ± 0.01
Oleic acid (C18:1n9c)	39.1 ± 0.01	15.2 ± 0.07	8.61 ± 0.05
Vaccenic acid (C18:1n7c)	-	1.38 ± 0.03	0.93 ± 0.02
*cis*-9.*trans*-12-octadecadienoic acid (C18:2n6ct)	0.11 ± 0.00	-	0.12 ± 0.00
*trans*-9,*cis*-12-octadecadienoic acid (C18:2n6tc)	0.10 ± 0.00	-	-
Linoleic acid (C18:2n6c)	9.05 ± 0.02	67.0 ± 0.05	52.8 ± 0.08
Arachidonic acid (C20:0)	0.33 ± 0.00	0.10 ± 0.00	0.83 ± 0.01
γ-Linolenic acid (C18:3n6)	-	-	4.26 ± 0.05
Gondoic acid (C20:1n9c)	0.15 ± 0.01	0.12 ± 0.01	0.38 ± 0.01
α-linolenic acid (C18:3n3)	0.16 ± 0.01	1.02 ± 0.01	17.47 ± 0.03
*cis*-11,14-eicosadienoic acid (C20:2n6)	-	-	1.52 ± 0.01
Behenic acid (C22:0)	-	-	0.33 ± 0.01
Lignoceric acid (C24:0)	-	-	0.13 ± 0.01
Calculated fatty acid content (%)			
SFAs	46.3 ± 0.01	10.4 ± 0.01	9.02 ± 0.01
MUFAs	39.4 ± 0.01	16.8 ± 0.03	10.0 ± 0.03
PUFAs	9.21 ± 0.02	68.0 ± 0.02	76.1 ± 0.03
omega-3	0.16 ± 0.01	1.02 ± 0.01	17.5 ± 0.02
omega-6	9.05 ± 0.02	67.0 ± 0.01	58.6 ± 0.02
omega-6/omega-3 ratio	56.6	65.7	3.3
TFAs	0.22 ± 0.00	-	0.12 ± 0.00

The results are presented as the mean ± SD (*n* = 3). SFAs, saturated fatty acids; MUFAs, monounsaturated fatty acids; PUFAs, polyunsaturated fatty acids; TFAs, trans fatty acids.

**Table 5 molecules-25-02921-t005:** Composition of the group-specific diets.

(g/100 g Diet)	Groups					
	LC	LPO	LHO	OC	OPO	OHO
Casein	20	20	20	20	20	20
DL-methionine	0.3	0.3	0.3	0.3	0.3	0.3
Palm oil	7	-	-	7	-	-
Poppy seed oil	-	7	-	-	7	-
Hemp seed oil	-	-	7	-	-	7
Corn starch	53	53	53	53	53	53
Saccharose	10	10	10	10	10	10
Cellulose	5	5	5	5	5	5
Mineral mix ^1^	3.5	3.5	3.5	3.5	3.5	3.5
Vitamin mix ^1^	1	1	1	1	1	1
Choline chloride	0.2	0.2	0.2	0.2	0.2	0.2

LC, lean rats fed a control diet; LPO, lean rats fed a diet with poppy seed oil; LHO, lean rats fed a diet with hemp seed oil; OC, obese rats fed a control diet; OPO, obese rats fed a diet with poppy seed oil; OHO, obese rats fed a diet with hemp seed oil. ^1^ Recommended for the AIN-93G diet [39].

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
