# Peer review of "Comparative Effects of Dietary Hemp and Poppy Seed Oil on Lipid Metabolism and the Antioxidant Status in Lean and Obese Zucker Rats"

_molecules, 2020, doi:10.3390/molecules25122921_

Round 1

Reviewer 1 Report

25-may-2020 Review for Molecules (ISSN 1420-3049)

Manuscript ID molecules-829554

Manuscript title:

Comparative effects of dietary hemp and poppy seed oil on lipid metabolism and antioxidant status in lean and obese Zucker rats

Authors:

Bartosz Fotschki , Paulina Opyd , Jerzy Juśkiewicz , Wiesław Wiczkowski , Adam Jurgoński

Comments:

The manuscript aim is interesting- to compare the effects of dietary inclusion of hemp seed oil (HO) and poppy seed oil (PO) on the lipid metabolism and antioxidant status of lean and genetically obese Zucker rats.

Comments:

The major concerns are about methodology and results.

From the method section is not clear, what were the control rats?

Were they also Zucker rats however lean, or other type, please make it clear?  

Why was in control chow palm oil, full of sasturated fatty acids? Authros mentioned that: „for 4 weeks, each group was fed a modified version of the semi-purified rodent diet recommended by Reeves (37). However, the AIN-93 rodent diets might be AIN-93G, formulated for growth, and AIN-93M, for maintenance. Major changes included substituting cornstarch for sucrose and soybean oil for corn oil and increasing the amount in order to supply both essential fatty acids (linoleic and linolenic).

Please add palm oil fatty acids profile to the table 4.

In the literature (Montoya et al. 2014 https://doi.org/10.1371/journal.pone.0095412) the composition of fatty acids of palm oil is: approximately 50% saturated fatty acids, with 44% palmitic acid (C16:0), 5% stearic acid (C18:0), and trace amounts of myristic acid (C14:0). The unsaturated fatty acids are approximately 40% oleic acid (C18:1) and 10% polyunsaturated linoleic acid (C18:2) and linolenic acid (C18:3)

The results in the table 1 are very confusing, according oil type the comparison between control, poppy and hemp seeds oils were together lean and obese rats, or only obese? Similarly comparison of rats phenotype lean vs. obese- this is comparison only rats on control diet? If so, please add always numbers of animals in each group.

The same table 2. It is not clear.

Please explain why there were no effects on body fat content, plasma lipid profile or lipid metabolism in the liver in the lean rats, HO and PO?

There is a very similar article published this year:

Paulina M Opyd 1, Adam Jurgoński 1, Bartosz Fotschki 1, Jerzy Juśkiewicz 1 Dietary Hemp Seeds More Effectively Attenuate Disorders in Genetically Obese Rats Than Their Lipid Fraction. J Nutr. 2020 Apr 10;nxaa081. doi: 10.1093/jn/nxaa081. PMID: 32275310 DOI: 10.1093/jn/nxaa081

Please comment it in the discussion add to references.

Both oils had unfavourable effects in obese rats as exhibited by the reduction of plasma HDL cholesterol levels. Additionally, poppy seed oil elevated plasma LDL cholesterol levels. The explanation in the discussion section is very weak.

Overall, I rate this paper low, and this type of manuscript should be rewritten.

Author Response

Comments:

The manuscript aim is interesting- to compare the effects of dietary inclusion of hemp seed oil (HO) and poppy seed oil (PO) on the lipid metabolism and antioxidant status of lean and genetically obese Zucker rats.

            We thank the Reviewer for comments.

Comments:

The major concerns are about methodology and results.

From the method section is not clear, what were the control rats?

Were they also Zucker rats however lean, or other type, please make it clear?  

In the experiment it was two control groups of Zucker rats, obese phenotype and lean phenotype. To clarify information about the rats and control groups in the “Material and Methods” section the following sentence “The rats were fed a control diet (C) containing palm oil as the sole source of fat (groups LC and OC; 7% of diet) or its modification where it was replaced with hemp seed oil (groups LHO and OHO) or poppy seed oil (groups LPO and OPO). “ was changed to “The Zucker rats from lean (LC) and obese (OC) control groups were fed a diet containing palm oil as the sole source of fat (7% of diet), whereas in the experimental groups the palm oil was replaced with the hemp seed oil (groups LHO and OHO) or poppy seed oil (groups LPO and OPO).” Moreover to make it more clear what type of rats were used in the experiment, in all Tables and Figures the name “Zucker” was added to the description. 

Why was in control chow palm oil, full of sasturated fatty acids?

The fatty acids profile of the palm oil and examined oils were determined using gas chromatography and method described in “Material and Methods” section. This analysis showed that palm oil contained considerable amounts of saturated and monounsaturated fatty acids. To make it more clear, the fatty acids profile of palm oil described under the Table 5 was shifted to the Table 4. Moreover, as it is described in the material and method section (lines 265-268), both palm oil in the control diet and its complete replacement with the tested oils were applied in order to obtain the most effective response of the organism to experimental factors, so that differences between poppy and hemp seed oils in terms of their health effects would be as clear as possible.   

Authros mentioned that: „for 4 weeks, each group was fed a modified version of the semi-purified rodent diet recommended by Reeves (37). However, the AIN-93 rodent diets might be AIN-93G, formulated for growth, and AIN-93M, for maintenance.

Under “Table 5” in the description it is specified that it was AIN-93G diet (line 279). To clarify what type of diet was used in the “Material and Methods” section specific type of diet “AIN-93G” was also added.

Major changes included substituting cornstarch for sucrose and soybean oil for corn oil and increasing the amount in order to supply both essential fatty acids (linoleic and linolenic).

The dietary level of sucrose and corn starch was not changed and it was at the same levels as recommended for AIN-93G diet. In the manuscript “Material and Method” section it is mentioned that experimental diets were similar in terms of dietary ingredients except for the type of oil (lines 257-258). The soybean oil was replaced by palm oil and examined oils due to reasons mentioned above and in the text (lines 265-268). The level of examined oils in all diets were in accordance to AIN-93G diet. We did not use corn oil to prepare diet.

Please add palm oil fatty acids profile to the table 4.

Fatty acid profile of palm oil was added To the table 4. 

In the literature (Montoya et al. 2014 https://doi.org/10.1371/journal.pone.0095412) the composition of fatty acids of palm oil is: approximately 50% saturated fatty acids, with 44% palmitic acid (C16:0), 5% stearic acid (C18:0), and trace amounts of myristic acid (C14:0). The unsaturated fatty acids are approximately 40% oleic acid (C18:1) and 10% polyunsaturated linoleic acid (C18:2) and linolenic acid (C18:3)

We thank Reviewer for the literature. The data are very similar to those obtained in our study (details in Table 4). However, we would like to emphasise that the paper is mainly focused at comparing fatty acid profile and health effects of hemp and poppy seed oils, because palm oil is already well studied.

The results in the table 1 are very confusing, according oil type the comparison between control, poppy and hemp seeds oils were together lean and obese rats, or only obese? Similarly comparison of rats phenotype lean vs. obese- this is comparison only rats on control diet? If so, please add always numbers of animals in each group.

The same table 2. It is not clear.

In the manuscript, a two-way ANOVA  was applied to present effects of examined oils, Zucker rat phenotype and the interaction between these investigated factors. This is a standard statistical procedure in such studies. To compare the effects of different types of oil in the statistical analysis we used lean and obese rats fed diet with palm, hemp and poppy oil, whereas to present effects of Zucker rat phenotypes we used obese and lean rats regardless to the type of oil. To clarify description of results in all Tables the number of animals in examined group was added.

Please explain why there were no effects on body fat content, plasma lipid profile or lipid metabolism in the liver in the lean rats, HO and PO?

The effects of examined oils on lean rats might be related to the homeostasis of the organism and a relative short time of feeding (4 weeks). It seems that when organism is not struggled with metabolic disorders the mechanisms regulating lipid metabolism and body fat content are more resistant to the changes in the diet especially when feeding period is only 4 weeks, therefore we speculate that there was no effects on body fat content, plasma lipid profile or lipid metabolism in the liver. However these experimental factors were sufficient to change parameters of antioxidant status in the lean rats. The following sentences were added to the discussion “Interestingly, changes in the blood lipid profile were not the case in healthy animals, which confirms our hypothesis that the health state of rats is an important factor in the modulation of lipid metabolism by the tested oils. The lean rats were not struggled with metabolic disorders,  therefore their organisms were probably able to maintain metabolic homeostasis easier despite dietary changes.”

There is a very similar article published this year:

Paulina M Opyd 1, Adam Jurgoński 1, Bartosz Fotschki 1, Jerzy Juśkiewicz 1 Dietary Hemp Seeds More Effectively Attenuate Disorders in Genetically Obese Rats Than Their Lipid Fraction. J Nutr. 2020 Apr 10;nxaa081. doi: 10.1093/jn/nxaa081. PMID: 32275310 DOI: 10.1093/jn/nxaa081

Please comment it in the discussion add to references.

The reference was added to the “Discussion” section as mentioned below.

Both oils had unfavourable effects in obese rats as exhibited by the reduction of plasma HDL cholesterol levels. Additionally, poppy seed oil elevated plasma LDL cholesterol levels. The explanation in the discussion section is very weak.

The effects of the examined oils on plasma lipid profile were partly related to different fatty acid composition. In the “Discussion” section, potential effects of different omega-6/3 fatty acids ratio as well as the presence of a trans fatty acid in hemp seed oil on lipid profile were described. To extend explanations, an information about the effects of rat phenotype on the lipid profile was added, as follows ”Another important factor in the modulation of lipid profile might had been a rat phenotype. In our previous study on obese Zucker rats, the reduction of plasma HDL cholesterol level after dietary supplementation with hemp seed oil was also observed [35]. Interestingly, changes in the blood lipid profile were not the case in healthy animals, which confirms our hypothesis that the health state of rats is an important factor in the modulation of lipid metabolism by the tested oils.”

Reviewer 2 Report

This study concluded that regular consumption of both oils increase the accumulation of essential fatty acids in the liver of healthy animals at the same time, do not cause  any adverse effects to the body, whereas in the genetically obese rats the effects of both dietary oils  on the lipid metabolism and antioxidant status are unequivocal and only partially beneficial.

General Comments~

  1. The results and discussions must describe more clear such as figure 1. 
  2. In discussion section, implications to table 3 for the association between patterns of weight change should focus on the results findings.
  3. Table 4. Fatty acid profile should include control diet.
  4. More exploration into these significant results.
  5. Figure 2 need to explain more clear.

Author Response

This study concluded that regular consumption of both oils increase the accumulation of essential fatty acids in the liver of healthy animals at the same time, do not cause  any adverse effects to the body, whereas in the genetically obese rats the effects of both dietary oils  on the lipid metabolism and antioxidant status are unequivocal and only partially beneficial.

            We thank the Reviewer for the comment.

General Comments~

  1. The results and discussions must describe more clear such as figure 1. 

To clarify description of the “Results” and “Discussion”, a table with P values from two-way ANOVA analysis presenting effects of examined oils, Zucker rat phenotype and the interaction between these investigated factors was added to Figure 1 and Figure 2, similarly as in tables.

  1. In discussion section, implications to table 3 for the association between patterns of weight change should focus on the results findings.

To highlight association between body weight change and significant results funded in Table 3, the following sentence was added to the “Discussion” section: “Previous studies have shown a clear association between increased body weight, obesity and higher synthesis of cholesterol [28]. Also in the present study the concentration of the liver cholesterol was considerably increased in the obese rats, however dieatary inclusion of hemp seed oil and, to a lesser extent, poppy seed oil lowered the concentration of liver cholesterol”.

  1. Table 4. Fatty acid profile should include control diet.

Fatty acids profile of the palm oil was added to Table 4. 

  1. More exploration into these significant results.

To extend explanation how hemp and poppy seed oil might modulate  lipid metabolism in the lean and obese Zucker rats to the “Discussion” section was added, as follows: “Another important factor in the modulation of lipid profile might had been a rat phenotype. In our previous study on obese Zucker rats, the reduction of plasma HDL cholesterol level after dietary supplementation with hemp seed oil was also observed [35]. Interestingly, changes in the blood lipid profile were not the case in healthy animals, which confirms our hypothesis that the health state of rats is an important factor in the modulation of lipid metabolism by the tested oils. The lean rats were not struggled with metabolic disorders, therefore their organisms were probably able to maintain metabolic homeostasis despite dietary changes. Nevertheless, the lipid metabolism between rats and humans differs, therefore human trials on the effects of dietary supplementation with hemp and poppy seed oils are necessary to confirm the results obtained here.”

  1. Figure 2 need to explain more clear.

Figure 2 was improved by adding P values from two-way ANOVA analysis presenting effects of examined oils, Zucker rat phenotype and the interaction between these investigated factors was added.

Reviewer 3 Report

The manuscript by Fotschki and co-authors, entitled:

“Comparative effects of dietary hemp and poppy seed oil on lipid metabolism and antioxidant status in lean and obese Zucker rats” provides very interesting and novel information on new potential  sources of plant oil that can be used to modulate the oil consumption together with the benefits of essential fatty acids from new plant sources.

In my opinion, this paper should be accepted for publication to “Molecules- Issn 1420-3049)

Author Response

The manuscript by Fotschki and co-authors, entitled:

“Comparative effects of dietary hemp and poppy seed oil on lipid metabolism and antioxidant status in lean and obese Zucker rats” provides very interesting and novel information on new potential  sources of plant oil that can be used to modulate the oil consumption together with the benefits of essential fatty acids from new plant sources.

In my opinion, this paper should be accepted for publication to “Molecules- Issn 1420-3049)

We thank the Reviewer for the comment.

Round 2

Reviewer 1 Report

Thanks to authors it si more clear

Reviewer 2 Report

All comments were revised.